# Biochemical Defense Responses in Red Rice Genotypes Possessing Differential Resistance to Brown Planthopper, *Nilaparvata lugens* (Stål)

**DOI:** 10.3390/insects14070632

**Published:** 2023-07-13

**Authors:** Prajna Pati, Mayabini Jena, Swarnali Bhattacharya, Santhosh Kumar Behera, Subhajit Pal, Raghu Shivappa, Tapamay Dhar

**Affiliations:** 1Faculty of Agriculture, Siksha ‘O’ Anusandhan Deemed University, Bhubaneswar 751030, Odisha, India; 2Department of Agricultural Entomology, Visva-Bharati University, Santiniketan 731236, West Bengal, India; 3Crop Protection Division, ICAR-National Rice Research Institute, Cuttack 753006, Odisha, India; 4Department of Agricultural Entomology, Indira Gandhi Krishi Viswa Vidyalaya (IGKV), Raipur 492012, Chhattisgarh, India; 5Regional Research Sub Station (OAZ), Uttar Banga Krishi Viswavidyalaya (UBKV), Mathurapur 732203, West Bengal, India; tdubkv@gmail.com

**Keywords:** brown planthopper, bio-chemicals, defense enzymes, host plant resistance, wild red rice genotypes

## Abstract

**Simple Summary:**

Rice, an imperative food crop, suffers from infestation by various insect pests. They cause considerable losses to rice production and quality. Among insect pests, the brown planthopper (*Nilaparvata lugens* Stål.) is a severe problem in the rice growing regions of the Indian sub-continent. Host-plant resistance is the safest way of managing this pest. We conducted a study to understand the differences in biochemical and defense enzyme activities in red rice genotypes, which showed different degrees of resistance to *N. lugens*. We experimented with net house condition. All the test genotypes were challenge infested with *N. lugens*. Cuttack population (*Biotype* 4). Changes in biochemical factors because of *N. lugens* feeding were assessed at 0, 24, 48, and 72 h. The results revealed significant differences in the quantity of total phenol and soluble protein, along with the activities of the defense enzymes such as peroxidase, polyphenol oxidase, catalase, and super-oxide dismutase among tested genotypes. The action of these defense-related enzymes was significantly higher in highly resistant genotypes, followed by resistant and moderately resistant genotypes. The crude silica content of all the genotypes showed a similar trend. In this experiment, we reported six highly resistant genotypes, namely Mata Meher, Manipuri Black, Hermonona, Sonahanan, Bavdi and Bacharya Khuta. The study might lead to the utilization of these lines in an *N. lugens* resistance breeding program.

**Abstract:**

The brown planthopper [*Nilaparvata lugens* (Stål.)] is one of the most destructive insect pests in all the rice-growing regions of the world. The pest is complicated to manage through the blanket application of chemical pesticides. The development of stable, durable *N. lugens*-resistant rice varieties is the most economical and efficient strategy to manage the pest. Landraces of red rice genotypes possess numerous nutritional and stress-resistant properties, though an exclusive study on the same is yet to be carried out. In the present study, we evaluated 28 red rice genotypes, along with two resistance checks and one susceptibility check, for their resistance to *N. lugens*. These promising lines revealed differential responses in the defense mechanism against the pest. The resistant accessions showed a greater accumulation of phenols, peroxidase, polyphenol oxidase, catalase, and superoxide dismutase under *N. lugens*-stressed conditions. However, the concentration of soluble proteins was substantially decreased in all the test genotypes. The concentration of crude silica was at maximum in highly resistant genotypes. Six red rice genotypes, namely Mata Meher, Manipuri Black, Hermonona, Sonahanan, Bavdi, and Bacharya Khuta fall under the highly resistant category, and can be utilized as valuable sources of resistance in breeding programs.

## 1. Introduction

Rice (*Oryza sativa* L.), being one of the world’s principal food crops, contributes immensely to the food security and economic sustainability of the global population [1]. Rice is cultivated by more than 60% of the small and marginal farmers of Asia and Africa, and in minimal quantities in America and Europe [2]. The production and productivity of rice increased by many folds after the introduction of high-yielding varieties during 1960s, bringing a green revolution to the country. Simultaneously, it has also brought new challenges to rice farming. The crop has numerous biotic and abiotic stresses at different phenological growth stages. Among the biotic stresses, insect pests are the primary obstacles to rice production, which can cause 30–100% crop losses under epidemic conditions [3]. The losses may account for about 20–80% of yield, and an overall annual economic loss of USD 300 Millions in Asia [4]. Though rice is facing problems from numerous insect pests, the brown planthopper, *Nilaparvata lugens* (Stål.) stands out as one of the destructive pests in South, South-East and East Asian rice growing regions [5]. *Nilaparvata lugens* damages the crop by sucking the sap from the phloem tissue. The pest, upon feeding, reduces the chlorophyll content and other biochemical constituents of plants. It hampers photosynthetic activity, and causes wilting and, ultimately, death of the plants [6]. Under epidemic conditions, “hopper burn” is the name given to an affected rice field that looks as if it were burned [7]. The nymphs and adult insects suck the sap by sitting at the base of the tillers and, most of the time, it goes unnoticed. Thus, symptoms only appear after wilting and, consequently, death of the plant starts. Apart from sucking the plant sap, the pest can also transmit viral diseases, such as grassy stunt, ragged stunt, and wilted stunt, accounting for significant crop loss [8]. The magnitude of crop damage and yield reductions depends on growing ecology, climatic conditions, and varietal susceptibility [9].

Management of *N. lugens* is challenging, as most of the insecticides became ineffective due to resistance development. The development of new biotypes is also very common [10]. Blanket application of chemical insecticides can seriously threaten native natural enemy populations and, ultimately, lead to a pest population resurgence. It also causes environmental pollution through pesticide residue in the soil, air, and water, which is detrimental to human and animal life [9]. Thus, the safest and most economical alternative for *N. lugens* management is the identification of new resistant donors and the development of resistant varieties, called host plant resistance. More than 39 resistant genes, or QTLs, have been discovered and successfully utilized in breeding programs of resistant varietal development. However, the changing pests’ biotypes, climatic conditions, and narrow genetic bases of the varieties have led to the breakdown of the resistance [10,11,12]. This has raised the serious necessity to screen more germplasm, including wild accessions, for a safe and durable resistance source [13]. Red rice/pigmented rice genotypes can provide some positive solutions. They possess nutritional and medicinal properties and have some valuable genes/QTL pools against biotic and abiotic stresses. Unfortunately, these genotypes are less explored for beneficial traits [14] because of their low yield potential and adaptability. The present investigation showed the biochemical basis of *N. lugens* resistance in promising red rice accessions identified as resistant to *N. lugens* in our previous study [5]. Understanding the biochemical and genetic factors responsible for resistance is the most critical aspect of managing *N. lugens*. Assessing the number of biochemical constituents expressed and contributing to *N. lugens* resistance will help to facilitate the resistance breeding program. Knowing the biochemical/defense enzymes induced before and after a pest’s attack, and their quantity, is necessary to conclude resistance or susceptibility [15]. Previous reports also showed that phenol compounds, crude silica, proteins, peroxidase, polyphenol oxidase, superoxide dismutase, and catalase contribute to *N. lugens* resistance [16]. This will provide useful information to develop broad-spectrum and durable, resistant varieties for eco-friendly management of brown planthoppers.

## 2. Material and Methods

A total of 28 promising red rice, accessions out of 215 initially screened genotypes for their resistance reaction against *N. lugens* [5], were selected to evaluate their possible biochemical defense response against the pest. Out of the twenty-eight accessions, four fall under the highly resistant (HR) category. Eleven and thirteen red rice accessions belong to the resistant (R) and the moderately resistant (MR) categories, respectively. Two resistant genotypes, viz. Salkathi and PTB-33, and one highly susceptible genotype, TN-1, used as national checks in the crop improvement programs, were included in the experiment, to compare the defense response of the red rice accession against the pest. Initially, healthy adult females/nymphs were collected from infested fields of an NRRI farm and released on healthy potted plants of the susceptible check TN-1 at a temperature of 30 ± 3 °C and 80 ± 5% relative humidity (RH). The plants were covered with cylindrical Mylar^R^ cages (45 × 15 cm^2^) to avoid insect escape. *N. lugens* culturing was performed as per the standard methodology on TN-1 plants [17]. Healthy seeds of red rice accessions were sown 5.0 × 1.0 cm^2^ apart in the plastic tray in rows at equal distances, with 20–25 rice seedlings of each genotype, along with check lines. The second group of instar *N. lugens* nymphs were released carefully in the screening trays containing 10-day-old seedlings, with 10 viruliferous nymphs per seedling. The plants were maintained at 28 ± 2 °C, 80 ± 5% relative humidity, and a 14:10 h light: dark photoperiod. Plant tissues from each red rice genotype were collected at 0, 24, 48, and 72 h after infestation (HAI) by the viruliferous nymphs for biochemical assay.

All the chemicals used in the present experiment were of molecular and analytical grade procured from *Sigma-Aldrich*, St. Louis, MO, USA. For spectrophotometrically recording the observations, we used a Thermo Scientific spectrophotometer (Model: Evolution 300). For confirming the *N. lugens* feeding on red rice, we used a trinocular stereo-zoom microscope (Nikon, Tokyo, Japan, SMZ745T, 2015) with 100× magnification.

### 2.1. Total Phenol

The phenol content was estimated as per the procedure given by Zeslinl and Zaken [18]. Briefly, 1 g of leaf sample from 20 to 25 seedlings was homogenized in 10 mL of 80% methanol. The mixture was agitated for 15 min at 70 °C. Exactly 1 mL of the methanolic extract was added to 5 mL of double-distilled water and 250 μL of 1 N Folin–Ciocalteau reagent (FCR:1 N). The resultant solution was kept at 25 °C for 3 min. The saturated solution of sodium bicarbonate and 1 mL of distilled water was added after three minutes, and the reaction mixture was incubated for 1 h at 25 °C. The absorption of the developed blue color was measured using a UV-visible spectrophotometer at 725 nm. The content of the total soluble phenols in each sample was estimated by plotting a standard curve obtained from a Folin–Ciocalteau reagent (FCR), with a phenol solution (C_6_H_6_OH) taken as standard. The total phenol content was expressed as catechol equivalents mg/g FW.

### 2.2. Assay of Defense Enzymes

Plant samples collected from *N. lugens*-infested and controlled rice seedlings (20–25 seedlings from each set) were immediately homogenized with liquid nitrogen into powder form. A total of 1 g of the powdered sample was extracted with 2 mL of the 0.1 M sodium phosphate buffer (pH 7.0). The resultant homogenate solution was centrifuged for 20 min at 10,000 rpm in a refrigerated centrifuge. The supernatant was collected in fresh tubes, and the pellet was discarded. The protein extract, thus prepared from red rice seedlings, was used for the estimation of defense enzymes [19] such as peroxidase (POD), polyphenol oxidase (PPO), superoxide dismutase (SOD), and catalase (CAT).

#### 2.2.1. Peroxidase (PO)

Peroxidase activity was assayed spectrophotometrically as per the methodology in [20]. A final volume of the reaction mixture consisted of 2.5 mL solution containing 0.25 per cent (*v*/*v*) guaiacol in 0.01 M sodium phosphate buffer at pH 6.0 and 0.1 M hydrogen peroxide (H_2_O_2_). Enzyme extract (100 µL) was added to initiate the reaction, followed by measuring the absorbance at 470 nm. Crude enzyme preparations were diluted at different concentrations in order to observe changes in absorbance units/min at 470 nm. Boiled enzyme extract was used as the blank. The activity of peroxidase in sample was measured as the increase in absorbance (δ) at 470 nm/min/g of fresh weight of tissue, and expressed as UA/g FW [20].

#### 2.2.2. Polyphenol Oxidase (PPO)

A total of 1 g of leaf tissue was homogenized in 2 mL of 0.1 M sodium phosphate buffer (pH: 6.5) at 4 °C. The homogenate was centrifuged at 20,000 rpm for 15 min at 4 °C in a refrigerated centrifuge. The pellet was discarded, and the supernatant was collected in a fresh tube. The supernatant served as enzyme source, and polyphenol oxidase (PPO) activity was determined as per the procedure in [21]. The reaction mixture consisted of 1.5 mL of 0.1 M sodium phosphate buffer (pH 6.5) and 200 μL of the enzyme extract. For initiating the reaction, 200 μL of 0.1 M catechol was added, and the change in absorbance was measured at 420 nm at different time intervals. The activity of PPO was calculated as a change in absorbance at 420 nm (δ)/min/g of tissue, and expressed as UA/g FW.

#### 2.2.3. Superoxide Dismutase (SOD)

In the cuvette, the reaction mixture containing 1.3 mL sodium carbonate buffer, 500 µL nitro blue tetrazolium (NBT) and 100 µL Triton X-100 was taken. The reaction was initiated by the addition of 100 µL hydroxylamine hydrochloride. After 2 min, 70 µL of the enzyme extract was added. The percentage inhibition in the rate of NBT reduction was recorded as an increase in absorbance at 540 nm. SOD activity was determined by following the standard procedure in [22].

The percent inhibition of NBT reduction was calculated as below:$$Y=\frac{Change\;in\;Abs./\min\left(blank\right)-Change\;in\;Abs./\min\left(Test\;Sample\right)}{Change\;in\;Abs./\min\left(Blank\right)}$$
where Y is the percentage of inhibition produced by 70 µL of the sample. Hence, 50% inhibition is produced by:$$Z=\frac{50\times 70}{Y}$$

One unit of the enzyme activity is defined as the enzyme concentration required for inhibiting the absorbance at 540 nm of chromogen production by 50% in one minute under the assay conditions. SOD activity was expressed as SA = mol UA/mg protein.

#### 2.2.4. Catalase (CAT)

Catalase activity was estimated as per the procedure in [23]. A total of 1 g of sample was homogenized in 10 mL of ice-cold 0.5 M sodium phosphate buffer (pH: 7.0) and centrifuged. The supernatant was used as an enzyme source. The reaction mixture consisted of 3 mL of hydrogen peroxide-phosphate buffer and 0.03 mL of enzyme extract. The rate of decomposition of H_2_O_2_ was followed by a decrease in absorbance at 240 nm, measured at 20 s intervals. The unit of the enzyme activity is calculated as the amount of enzyme required to liberate half the peroxide oxygen from H_2_O_2_ and calculated from the following equation:$$Unit\;Activity\;(units/\min FW/g)=\frac{Change\;in\;Abs./\min\times Total\;Volume(mL)}{Ext.Co-efficient\times Vol.\;of\;the\;sample}$$
where the extinction coefficient = 6.93 × 10^−3^ mM^−1^ cm^−1^.
$$Specific\;Activity\;(Mol\frac{UA}{mg}protein)=\frac{Unit\;activity\;(\frac{U/mFW}{g})}{Content\;of\;Protein\;(FW\frac{g}{mg})}$$

CAT activity is expressed as SA = mol UA/mg protein.

### 2.3. Soluble Protein

Soluble protein concentration was determined by following the standard method in [24]. A total of 0.1 mL of the sample and standard were pipetted into a series of test tubes. The volume of test tubes was made up to 1.0 mL with distilled water. A tube with 1.0 mL of distilled water served as the blank. A total of 5.0 mL of reagent C (2% Sodium carbonate in 0.1 N sodium hydroxide + 0.5% copper sulphate in 1% potassium sodium tartrate) was added to each tube. After mixing it properly, it was allowed to stand for 10 min. Then, 0.5 mL of reagent D (FC Reagent protein solution + 50 mg of BSA) was added, mixed well, and incubated at room temperature in the dark for 30 min, until the sample mixture developed a blue color. The absorbance was recorded at 550 nm. A graph of absorbance vs. concentration for standard solutions of proteins was plotted, and the amount of protein in the samples was calculated from the graph. The quantity of proteins was expressed as mg/g FW.

### 2.4. Crude Silica

Crude silica was estimated by following standard protocol with required modifications, as seen in [25]. The leaf sample was dried in an oven at 70 °C for 7 days. It was ground to prepare a fine powder, and sieved through 60 mesh sieves. The sample was then dried at 60 °C for 48 h. From this, 100 mg of sample was weighed in a 100 mL polythene graduating tube, added to 30 mL of 50% NaOH, and vortexed. The reaction mixture was transferred to 50 mL volumetric flask, and the volume was made by adding distilled water. Thereafter, 1 mL of sample was taken in a 50 mL volumetric flask, and 30 mL of 20% acetic acid and 10 mL of ammonium molybdate solution were added to it. The solution was shaken well and kept at room temperature for 50 min. To the above solution, 5 mL of 20% tartaric acid and 1 mL of reducing agent were added. The volume was then made up to 50 mL with double-distilled water. The solution was kept for 30 min. The reaction mixture was then used for crude silica estimation by measuring the absorbance at 650 nm. The crude silica content was expressed in percentage.

### 2.5. Statistical Analysis

The data obtained from different experiments of biochemical analysis was analyzed by using completely randomized design (RBD). The data was subjected to the square root of X + 1 transformation before statistical analysis. The analysis was carried out using SAS 9.1 online software (IASRI, New Delhi, India). The data was analyzed for levels of significance (*p* < 0.001) of the main treatments, and their interactions with other factors (unbalanced confounded factorial design) were calculated by analysis of variance after testing for normality and variance homogeneity. The means were separated by Tukey’s test for significant differences between treatments. Principal component analysis (PCA) and cluster analysis (UPGMA) were performed with a variance–covariance matrix, and a comparison using Palaeontological Statistics software (PAST v.4.03) was disregarded. Graphical representation of the data was drawn using R software 4.3.1.

## 3. Results

We performed a quantitative analysis of each of the biochemical parameters to assess the variations of these parameters among the red rice accessions at a particular time point in *N. lugens*-stressed conditions. We also consider the interaction effects of three factors, namely genotypes (factor 1), defense enzymes (factor 2), and time series (factor 3). All interactions between the factors showed significant differences (*p* < 0.0001) among each other (Appendix A).

### 3.1. Total Phenols

Significant variation in phenol content was observed among the red rice accessions at different hours after infestation by *N. lugens* (*p* < 0.001). Phenol content of all the genotypes increased significantly over time. High level of phenolics was noticed in all four highly-resistant genotypes and two of the resistant genotypes. The lowest quantity of phenolics was recorded in the highly susceptible check genotype TN 1 (Figure 1 and Appendix A). The phenol content reached its peak concentration after 48 h in all the test genotypes, with the maximum concentration in Salkathi (71.32 mg/g FW), Mata Meher (65.98 mg/gFW), Hermonona (64.34 mg/g FW) and Bacharya Khuta (63.15 mg/g FW). It clearly indicates increased phenol accumulation in resistant genotypes in response to *N. lugens* feeding. The phenol content started declining rapidly after 72 h in susceptible check TN-1.

### 3.2. Peroxidase (POD)

Peroxidase enzymes showed similar trends, where significant variation (*p* < 0.001) was observed among the genotypes, with a peak period of activity at 48 h after *N. lugens* infestation (Figure 2 and Appendix A). The HR genotype Mata Meher showed the highest POD activity (6.30 UA/g FW), followed by another HR genotype, Manipuri Black (6.15 UA/g FW). The lowest concentration was recorded in HS check TN 1 (3.62 UA/g FW). The peroxidase content started declining at 72 h of *N. lugens* infestation, and more rapid decline was noticed in the susceptible check genotype compared to resistant genotypes.

### 3.3. Polyphenol Oxidase (PPO)

*Nilaparvata lugens*-infested red rice genotypes showed a significant variation (*p* < 0.001) in PPO activity when compared to same plants before *N. lugens* feeding (Figure 3 and Appendix A). PPO activity increased over the time of infestation and reached its peak at 48 h. The activity then gradually decreased after 72 h of feeding. The maximum PPO content was recorded in HR accession Mata Meher (12.36 UA/g FW) at 48 h, followed by HR accession Sonahanan (11.87 UA/g FW). The HS check TN 1 here also possessed the lowest concentration of PPO (6.74 UA/g FW).

### 3.4. Superoxide Dismutase

The SOD activity also significantly changed over time among the rice genotypes (*p* < 0.001), with its peak at 48 h of feeding (Figure 4 and Appendix A). R genotype Bacharya Khuta had the highest SOD activity (5.38 mol UA/mg protein), which was on par with R genotype Bavdi (5.33 mol UA/mg protein) and the other HR genotypes Mata Meher (5.37 mol UA/mg protein), Sonahanan (5.53 mol UA/mg protein), Hermonona (5.22 mol UA/mg protein), Manipuri Black (5.37 mol UA/mg protein), Salkathi (5.08 mol UA/mg protein), and PTB-33 (5.23 mol UA/mg protein). HS check, TN-1 recorded the least enzyme activity in all feeding stages of *N. lugens*.

### 3.5. Catalase (CAT)

The concentration of catalase (CAT) in all the tested genotypes differed significantly after *N. lugens* feeding (*p* < 0.001). However, the highest activity of CAT was observed 24 h after insect feeding (Figure 5 and Appendix A) instead of 48 h, as evident from other defense enzymes. The resistant genotype Bavdi showed the highest activity (45.55 mol UA/mg Protein) followed by Bachaya Khuta (42.01 mol UA/mg protein), Maria dhan-2 (40.48 mol UA/mg protein FW) and Mata Meher (40.04 mol UA/mg protein). In all test red rice genotypes, the catalase (CAT) activity started declining after 24 h. The lowest enzyme activity was found in the susceptible check genotype TN 1.

### 3.6. Soluble Proteins

We observed significant changes in soluble protein content (*p* < 0.001) in all the test genotypes after *N. lugens* feeding (Figure 6 and Appendix A). Initially, the moderately resistant genotype Meghi recorded the highest total soluble protein content (18.47 mg/g FW). The most negligible concentration was observed in highly-resistant genotype Manipuri Black (6.67 mg/g FW) and Mata Meher (7.25 mg/g FW), while a considerable quantity of soluble protein was noticed in the highly susceptible cultivar TN-1 (15.25 mg/g FW). Soluble protein concentration continued to decline due to feeding by *N. lugens*. All the genotypes recorded low soluble protein levels at 72 h of feeding.

### 3.7. Crude Silica (Si)

Significant variations (*p* < 0.001) in the percent crude silica content was observed in all the tested red rice genotypes after 24 h of *N. lugen* feeding (Figure 7 and Appendix A). The highly-resistant genotype Mata Meher had the highest silica content (8.42%), statistically on par with Manipuri Black (8.08%). Notably, other highly resistant genotypes, namely Salkathi (7.42%), Sonahanan (7.25%) and PTB-33 (6.92%), recorded higher crude silica content compared to the susceptible check genotype TN-1 (2.48%). Hence, the silicification of the rice sheath was highest in resistant genotypes compared to susceptible ones, indicating its essential role in *N. lugens* resistance.

Critical introspection of the data revealed more induction of phenol and other oxidative defense enzymes in the resistant genotypes over the susceptible genotype under biotic stress conditions due to *N. lugens* feeding during the peak period of activity (Figure 8). The percentage induction of phenol and peroxidase in all the resistant genotypes was higher than the highly-susceptible check TN-1. We recorded a 220.25% and 236.15% higher phenolic content and peroxidase activity in Bavdi and Manipuri Black, respectively, over the susceptible genotype TN-1 during the peak period of activity at 48 h after insect feeding. The percentage induction of catalase in its peak period of activity was more in all the test genotypes, except Bavdi, over the susceptible check TN 1. In the case of PPO, we notice a similar trend in five genotypes. The percentage induction of SOD over the susceptible check was recorded only in the genotype Sonahanan.

More interestingly, all four HR genotypes showed enhanced expression of POD, compared to the two resistant checks, during 48 h of insect feeding. Three of them had more up-regulation of catalase enzyme over the resistant checks at the peak period of activity, i.e., 24 h after insect feeding. At the same time, two of them showed more induction of phenolics over the resistant checks.

Two resistant genotypes, viz. Bavdi and Bacharya Khuta, induced more phenolics over the resistant checks 48 h after insect feeding. The Bavdi genotype only showed more up-regulation of PPO over the resistant checks during the peak period of activity of the enzyme under *N. lugens*-stressed conditions.

### 3.8. Principal Component (PCA) and Cluster Analysis

We performed a principal component analysis (PCA), using Past.4 software to identify the grouping patterns among the red rice genotypes, based on biochemical and defense enzyme constituents. The first two principal components accounted for 80.21% and 12.83% of variance (Figure 9). Four components with eigenvalues of more than 1 (Figure 10) were recorded (PC1: 80.21%, PC2: 12.83%, PC3: 4.81 and PC4: 1.29).

Allocation of the variables to their principal components is a subjective approach, and was performed based on their PC scores. Principal components that were not correlated with each other indicated that different principal components were responsible for explaining the variability in different variables. Clear positive coefficients between red rice genotypes and biochemical constituents such as phenol, POD, PPO, SOD, CAT, and Silica were observed, while a negative coefficient related to total soluble protein content was observed (Figure 11).

Phylogenetic grouping of the red rice genotypes (Figure 12), using unweighted pair group method with arithmetic mean (UPGMA), showed a clear clustering of all the resistant genotypes in one cluster (Cluster-I) and susceptible check TN-1 in a separate cluster (Cluster-II) with only susceptible check TN-1.

In Cluster-I, we observed a clear grouping of highly resistant genotypes, namely Mata Meher, PTB-33, Manipuri Black, Sonahanan, Hermonona and Salkati. The cluster of resistant genotypes through PCA almost corroborates with the outcome of the mass screening method. Two additional genotypes, Bacharya Khuta and Bavdi, which showed moderate immune reactions in the mass screening, come under the highly resistant group. A heat map also shows a clear separation of the genotypes according to their resistance level. Here, the resistant genotypes designated by different colors indicate their level of resistance to *N. lugens* (Figure 13). The defensive enzymes and phenolics in the plant systems played a crucial role in the resistance mechanism, corroborating with Figure 8.

## 4. Discussion

The damage by insect pests accounts for 10% of rice crop field losses [4,26]. Among them, the brown planthopper (*N. lugens*) alone stands out as one of the most destructive pests. The pest is a phloem-feeding insect distributed around tropical rice-growing regions, including India. Several epidemic outbreaks have been reported in different states of the country [3]. Host plant resistance is an eco-friendly and economical approach to overcome the pest problem. The plant’s immune system, and various biochemical and defense-related genes, can protect the rice plants from *N. lugens* attack [27]. Rice plants have developed a sophisticated defense system against *N. lugens* in the long-term battle between the two.

The crop can defend against *N. lugens* attack by producing various defense enzymes/proteins or chemicals, which reduce insects’ digestion and absorption of nutrients [28]. Biochemical factors, such as phenols and OD phenols, can significantly contribute to plant defense [16,29]. A wide range of defense mechanism systems pre-existing in higher plants, including antioxidants and protective enzymes, help them against various biotic stresses [30]. The enhanced activities of antioxidative enzymes, such as superoxide dismutase (SOD), peroxidase (POD), and catalase (CAT), are the components of defense against membrane lipid peroxidation caused by reactive oxygen species, ROS [31]. Peroxidase (POD), polyphenol oxidase (PPO), and catalase (CAT) are involved in the biosynthesis of secondary metabolic compounds, such as phytoalexins, phenols, and lignin, which contribute to the resistance against *N. lugens* [32,33,34]. Understanding the biochemical mechanisms, along with genetic factors, contributing to resistance in rice is paramount to managing the *N. lugens* population, and will felicitate the resistance breeding program [15]. Silicon, now recorded as a “beneficial substance” or “quasi-essential” element, may mediate defense against insect herbivores in several ways, by activating mechanical barriers and biochemical defense systems [35,36]. Our study observed a more significant accumulation of phenol content in highly-resistant genotypes compared to the susceptible check. The higher accumulation of phenols through the shikimic acid pathway in the plant system acts as a growth inhibitor to insect pests. The results agree with previous studies, such as Srinivasan and Uthamasamy [37], who recorded higher accumulation of phenolic compounds in tomatoes against *Helicoverpa armigera* and *Bamisia tabaci*. Some rice accessions, such as ADT45 and PTB 33, showed higher induction of phenols in response to *N. lugens* attack [38].

Peroxidase activity has a direct relation with disease and insect pest resistance in several plants [39,40,41,42]. Peroxidase, involved in several physiological functions of plants, contributes to the resistance against an invading pest or pathogen. Peroxidase oxidizes hydroxyl cinnamyl alcohol into a free radical intermediate. It also plays a role in the oxidation and polymerization of phenolics, cross-linking polysaccharides, and extension monomers. It has a very crucial role in lignification. It shows hypersensitive reactions, and adverse effects on food digestibility and protein availability to sucking pests [39]. Hence, early and increased expression of peroxidase has a role in lignification process that can protect the host plants against sucking insects. In the present investigation, all resistant red rice genotypes revealed an enhanced expression of this essential oxidative enzyme in *N. lugens*-stressed conditions. Such an expression of peroxidase might have contributed to the resistance phenomenon of the red rice accessions. Our findings corroborate the results of others on *N. lugens* and *Cnaphalocrosis medinalis* G. in rice [43,44,45].

Polyphenol oxidase (PPO) activity increased in all the resistant genotypes up to 48 h after *N. lugens* infestation, with a more significant expression in HR genotypes. PPO reduces the nutritional quality of infested plants by converting soluble phenolic compounds into quinines, and ultimately prevents the digestion of proteins in insects, thus making rice plants non-preferable to *N. lugens*. The PPOs are well-studied oxidative enzymes. They are widely distributed in plant systems, and their effects on damaged plant tissues have been known for many years [35,46]. Jasmonate-inducible proteins (JIPs) that have a confirmed or proposed role in post-ingestive defense include polyphenol oxidase, arginase, leu amino peptidase A (LAP-A), lipoxygenase, and a battery of PIs [47]. Similar results were reported by two reporters [48,49] against tomato fruit borer (*Helicoverpa armigera*) and maize cob borer (*Helicoverpa zea*), where increased activities of PPOs have affected the growth and development of these lepidopteran pests. Similarly, [44] reported that timely and increased expression of PPOs in PTB-33, ASD-7, ADT-45, CO-43, and KAU-1661 contribute significantly to their resistance. Increased activity of PPOs reduces the growth and development of *Cnaphalocrosis medinalis* in rice plants [50].

Superoxide dismutase (SOD) acts as the first line of defense against pests, contributing to the cell wall structures to increase resistance [51]. It is a major component of insects’ antioxidant enzyme system, which converts superoxide radicles into oxygen and hydrogen peroxide (H_2_O_2_). Generally, high SOD activity leads to lower membrane lipid peroxidation. Such a defense response is an important factor in host plant resistance against insect pests. The success of plant defense responses depends on quick recognition and immediate action against insect attack [52]. The role of SOD is well-studied in cucumber plants against *Bemisia tabaci* [53]. Here, the induction of activities of SOD may contribute to the bio-protection of cucumber plants against the white fly. Similarly, enhanced activities of SOD and CAT were observed in response to aphid (*Acyrthosiphon pisum* Harris) infestation in peas [54]. In our study, the SOD activity increased upon *N. lugens* infestation and peaked at 48 h. However, SOD is less expressive in all the test genotypes except one HR genotype, Sonahanan, as compared to the highly susceptible check TN-1. Some reports revealed that superoxide dismutase activity decreased in rice plants infested with three insect pests, such as yellow stem borer and *N. lugens*, compared to the control plants [38]. Antioxidant-related gene *SOD* shows decreased relative expression in rice plants on *N. lugens* attack [55]. Further, SOD activity did not influence the level of *Myzus persicae* resistance in tobacco [56].

Catalase is one of the most crucial defense enzymes in plant systems. It is a prime H_2_O_2_-scavenging enzyme, and removes excessive H_2_O_2_ generated during the developmental stage, or by environmental stimuli through water and oxygen [57]. Increased levels of CAT are involved in plants’ cell wall resistance [58], and act as a signal for the induction of defense genes [59]. In our study, CAT activity slightly increased in the *N. lugens*-infested red rice genotypes up to 24 h, then reduced significantly. Some previous workers recorded similar results. Decreased activity of CAT was observed in a Russian wheat aphid infestation [60]. The homopteran insects feed from the contents of vascular tissue by inserting a stylet between the overlying cells, thus limiting the cell damage, and minimizing induction of the wound response. Contrary to this observation, Usha Rani and Jyothsna [38] demonstrated that YSB- and LR-infested rice plants showed increased CAT activity in response to the extent of cell damage. Similarly, increased CAT activities in soybeans were observed against a *Helicoverpa zea* infestation [61].

Increased expression of defense enzymes and their synergistic effects plays a very crucial role in the resistance mechanism of rice plants against *N. lugens* [62,63], which also corroborates our findings.

Soluble protein in host plants is the primary source of amino acids, and an indicator of food quality for herbivores. A significant decrease in protein content due to herbivore attack was reported [64]. We also noticed significantly reduced soluble protein content in all the test genotypes with *N. lugens* feeding in time intervals. *N. lugens* and leaf folder infestations reduce rice plants’ photosynthetic activity, thus decreasing soluble protein content [65,66]. White-backed plant hoppers and *N. lugens*, upon feeding, decline the leaf protein content of rice by 40–60%, compared to control plants [67]. Jayasimha et al. [68] reported a higher total soluble protein content in susceptible cultivar TN-1 than in resistant PTB-33. With all these observations, we can conclude that protein content is reduced at different degrees based on the resistance level of the insects.

As an important element, silicon contributes significantly to the plant’s defense against a herbivore attack in Poaceae plants [69]. Silicon increases the plant’s physical barrier and alters herbivorous feeding behaviors by enhancing the plant’s defense. Physical defense is associated with accumulating absorbed silicon in the epidermal tissue as a mechanical barrier in leaf epidermis cells, increasing hardness that causes wear to insect mandibles, and reducing digestibility [70]. Soluble silicon induces a biochemical defense against insect pests’ attacks through enhanced production of defensive enzymes and phenolic compounds [71,72,73]. We observed higher crude silica content in resistant red rice genotypes. Previous researchers noticed increased abrasiveness and rigidity of plant tissues resulting from increased silicification of rice sheaths with a silicon addition, thus making it difficult for *N. lugens* feeding [69,74].

## 5. Conclusions

Researchers are undertaking the identification of a stable resistance source against brown planthoppers, and novel sources with different genes/QTLs have been reported from wild and cultivated species. However, resistance breakdowns in *N. lugens* are widespread. We screened wild red rice genotypes for their resistance against *N. lugens*, and their underlying mechanisms. We found six red rice genotypes imparting resistance against *N. lugens*. Highly-resistant red rice accessions under *N. LUGENS*-stressed conditions showed an enhanced expression of defense enzymes. Further, the resistant genotypes had more phenol and silica content. The higher expression of these compounds might interfere with *N. lugens*’ further growth and development. These enzymes might reduce the nymphal preference, fecundity, feeding rate, survival, growth index, and population buildup of *N. lugens*. These results help to formulate an efficient strategy for integrated pest management of *N. lugens* in rice, and the development of resistant varieties through advanced breeding programs.

## Figures and Tables

**Figure 1 insects-14-00632-f001:**
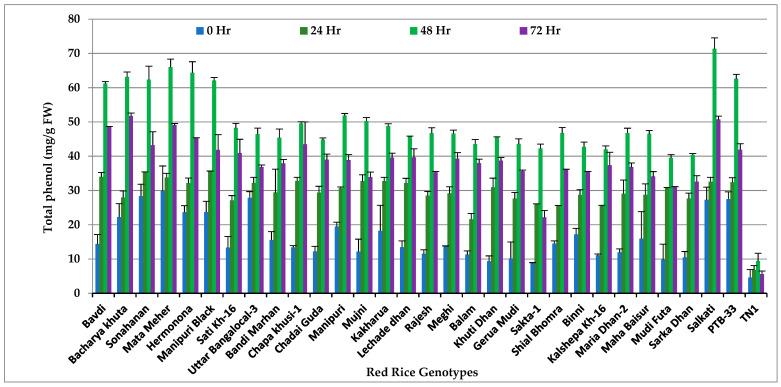
Change in total phenol (mg/g FW) in red rice genotypes due to *N. lugens* infestation.

**Figure 2 insects-14-00632-f002:**
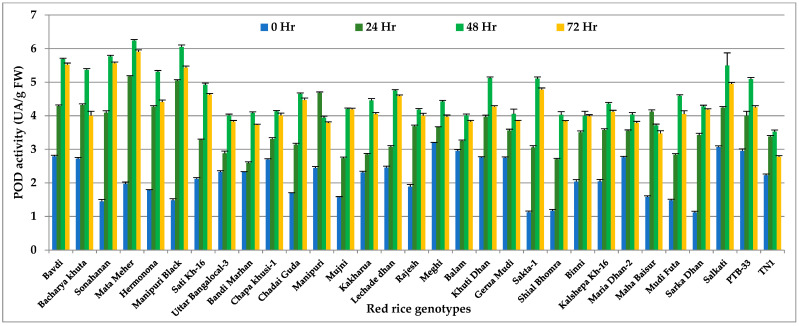
Change in POD activity (UA/g FW) in red rice genotypes due to *N. lugens* infestation.

**Figure 3 insects-14-00632-f003:**
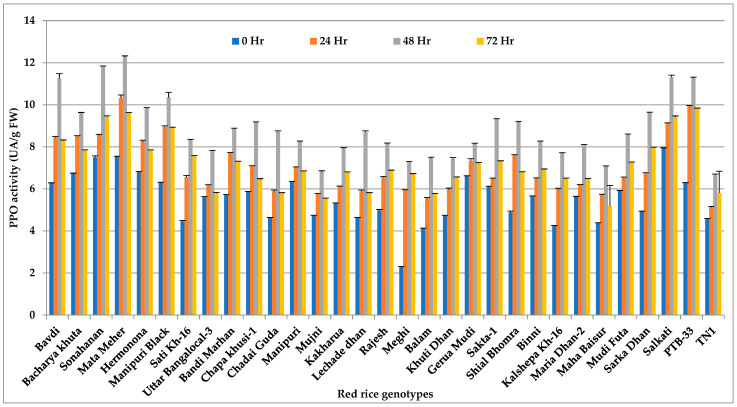
Change in PPO activity (UA/g FW) in red rice genotypes due to *N. lugens* infestation.

**Figure 4 insects-14-00632-f004:**
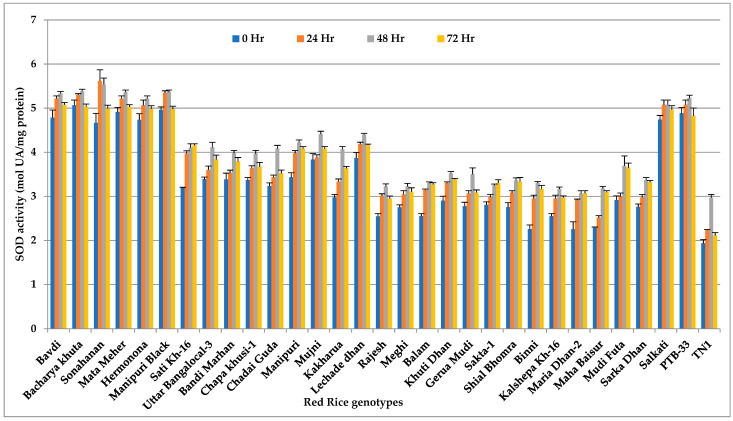
Change in SOD activity (mol UA/mg Protein) in red rice genotypes due to *N. lugens* infestation.

**Figure 5 insects-14-00632-f005:**
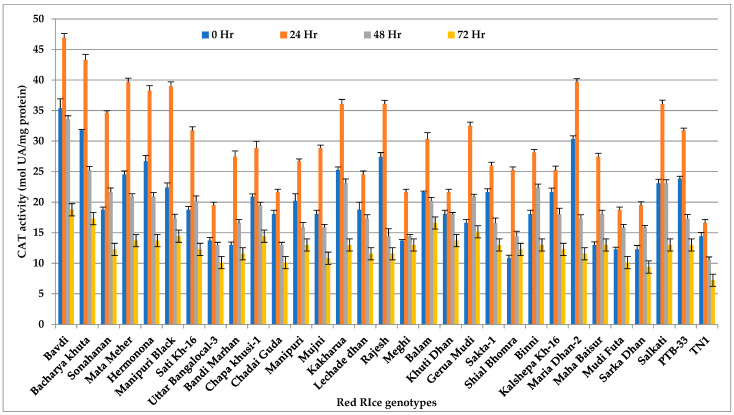
Change in CAT activity (mol UA/mg protein) in red rice genotypes due to *N. lugens* infestation.

**Figure 6 insects-14-00632-f006:**
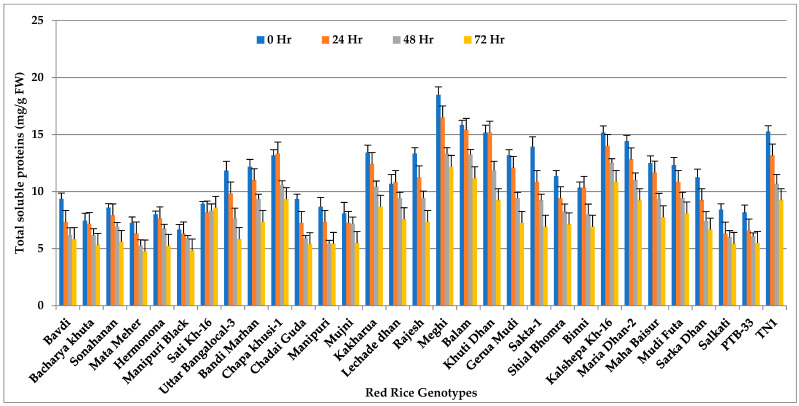
Total soluble proteins (mg/g FW) in red rice genotypes due to *N. lugens* infestation.

**Figure 7 insects-14-00632-f007:**
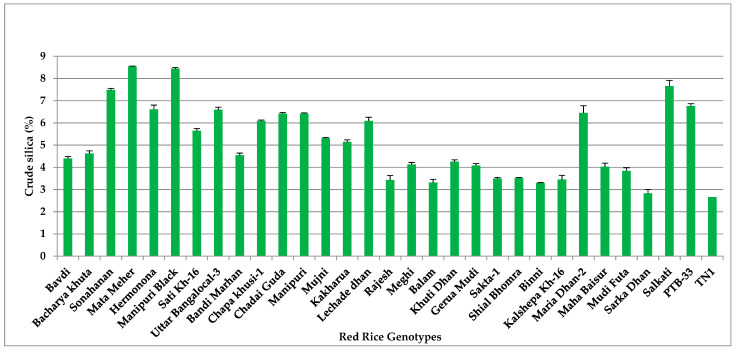
Crude silica (%) in red rice genotypes due to *N. lugens* infestation.

**Figure 8 insects-14-00632-f008:**
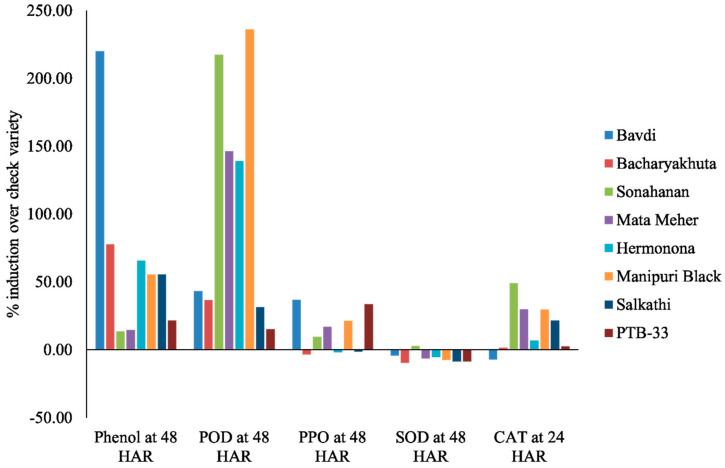
Over-expression of phenolics and defense enzymes in resistant genotypes over susceptible genotype during peak period of activity.

**Figure 9 insects-14-00632-f009:**
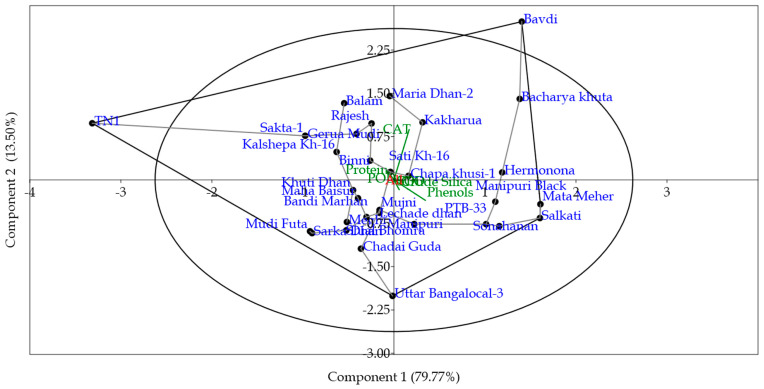
Principal component analysis (PCA) plot of various biochemical traits in the red rice genotypes under *N. lugens* stress: a grouping of the variables into two principal components.

**Figure 10 insects-14-00632-f010:**
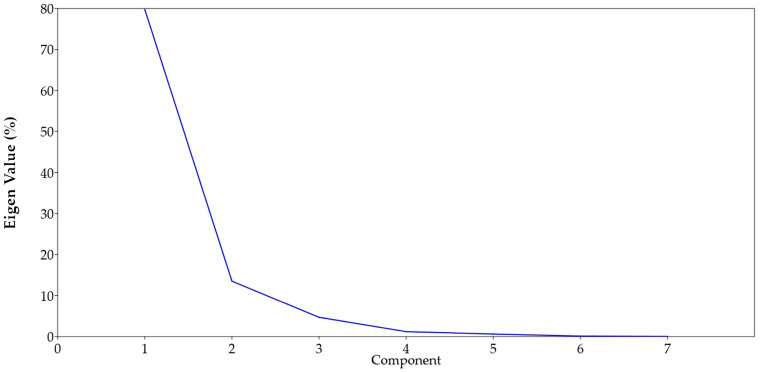
Four components with their eigen values.

**Figure 11 insects-14-00632-f011:**
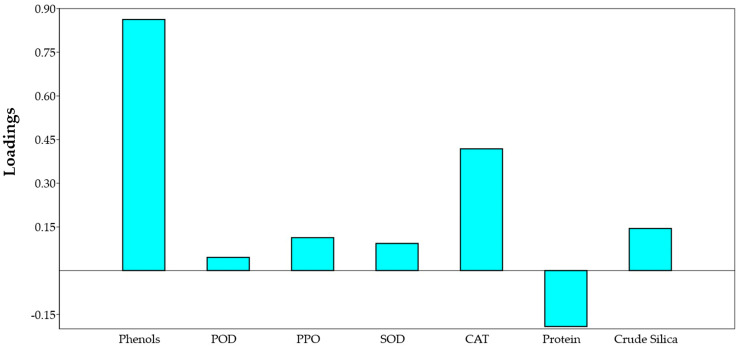
Loading diagram of the relations among biochemical components with rice genotypes.

**Figure 12 insects-14-00632-f012:**
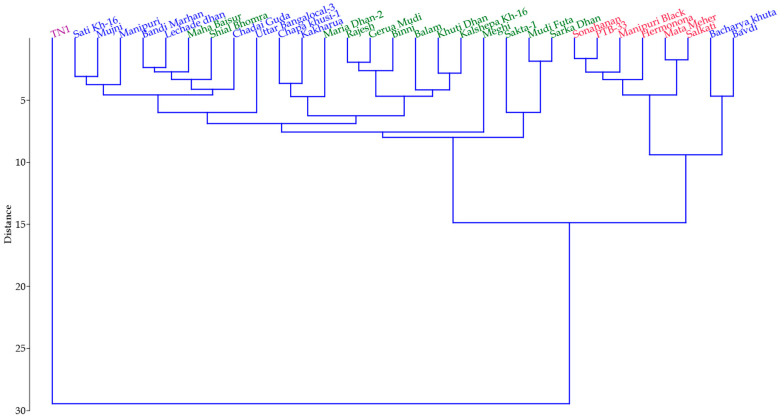
Grouping of the red rice accessions, using UPGMA, into different clusters based on biochemical components.

**Figure 13 insects-14-00632-f013:**
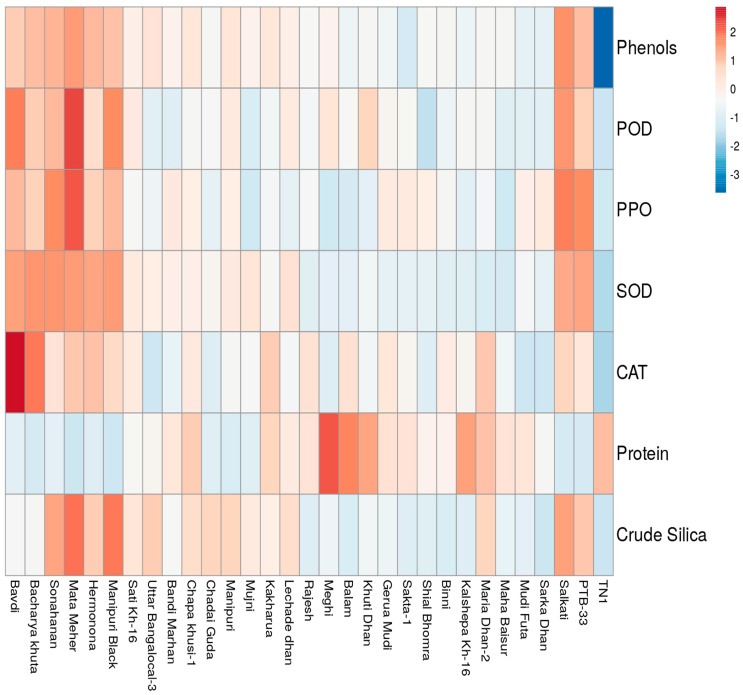
Heat map of the defense enzymes showing clear separation based on their resistance grades.

## Data Availability

All the data sets generated for this research are included in the manuscript or Appendix A.

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
