# Peer review of "Biochemical Defense Responses in Red Rice Genotypes Possessing Differential Resistance to Brown Planthopper, Nilaparvata lugens (Stål)"

_insects, 2023, doi:10.3390/insects14070632_

Round 1
Reviewer 1 Report
Overally, this is an interestingly study about the resistance of different rice genotypes to the brown planthopper, a severe rice pest. This study investigated a lot of genotypes and resistance parameters. The results are valid and important. I just have a few concerns about the statistical analyses. It would be better to use different letters to indicate significant differences in different time points for each genotype. To compare among genotypes, it would be better to calculate inhibition/induction indexes and evaluate effect of planthopper attack on each resistance parameter.
Author Response
Please see the attachment.
with regards,
Dr Swarnali Bharttacharya

Reviewer 2 Report
The paper by Pati et al examines resistance to N. lugens feedings in red rice cultivars for screening the genetic resources of host plant resistance. The authors found the candidates of the resistance cultivars of red rice. However, I think that authors failed to the statistical analysis of biochemical defenses data in screening tests. Authors should conduct the three-way ANOVA rather than one-way ANOVA with Tukey’ method. Authors used same data for analyzing the one-way ANOVA as a quantitative analysis and UPGMA as a qualitative analysis. I think that authors should focus their statistical methods on either qualitative or quantitative analysis of biochemical data in red rice.
All figures have low resolution, so the text is not clear in figures; please upload the figures in pdf vector format instead of png or jpeg raster format. Especially, in figures of UPGMA and PCLA analysis, I don’t determine whether the description of analysis results adequately illustrate the figures or not. Authors should provide readers high resolution figures. Please re-draw the figures in R or edit the figures in Illustrator.
Please fix the mistake in terminology and scientific names.
Author Response
Please see the attachment.
with regards,
Dr Swarnali Bhattacharya

Reviewer 3 Report
The manuscript ‘ Biochemical Defense Responses in Red Rice Genotypes Pos3 sessing Differential Resistance to Brown Planthopper, Nilapar4 vata lugens (Stal.), describes the differences in chemical and peroxidase content in several red rice genotypes. The manuscript seems to have great significance, however, the quality of the figures is too low and makes it difficult to understand.
Minor suggestions:
It would be great to see any marker gene expression (related to catalase/ peroxidases) in the resistant genotypes compared to others.
Line 20. … undertaken to understand the differences….
Line 30. This study might lead to the utilization of these lines…….
Line 32: remove only
Minor changes are required.
Author Response
Please see the attachments.
with regards,
Dr Swarnali Bhattacharya

Reviewer 4 Report
My overall comments are as follow:
1. The results are overall clearly presented.
2. The unit of enzyme activity should be unified. Change the unit of enzyme activity to “U/g FW (fresh weigh)”
3. Figures are not clear. Please use high resolution figures in the manuscript.
4. Check the symbols. A lot of symbol mistakes should be modified.
Line 4: Please change “Nilaparvata lugens (Stal.)” to “Nilaparvata lugens (Stål)”
Line 26: “The activity was significantly higher” this statement is not clear enough. Maybe you mean “The activity of these defense-related enzymes was significantly higher”
Line 31: Please change “Nilaparvata lugens Stal.” to “Nilaparvata lugens (Stål)”
Line 111: the degree symbol is incorrect. Change it to °C.
Line 112: the size of the cages
Line 114: Please change “5.0 X 1.0 cm” to “5.0 × 1.0 cm2”
Line 126: How many replicates (how many seedlings) were used in the extraction of total phenol for each variety? 20-25 seedlings?
Line 138: how many infested seedlings and uninfested control were used in the assay of defense enzymes?
Line 174, 179, 194, and 199, please use Insert > Equation function in WORD to create these formulas.
Line 239, 251, 262, 286, 297, 308: “P>0.001”? I guess it should be “P<0.001” as you mentioned in the supplementary file.
Line 251: “P>0.001”?
Line 254: “highest POD content” should be “highest POD activity”
Minor editing of English language required.
Author Response

(The authors gave the same response as above.)

Round 2
Reviewer 1 Report
The manuscript was significantly improved, and I suggest accepting the MS. Please note that some figures are still can be improved (e.g., Fig. 11)
Reviewer 2 Report
Authors tried to improve a manuscript following reviewer's comment.
I think that authors additionally improve the manuscript below two points for the acceptance.
1.  BPH is changed to N. lugens in a whole manuscript.
2. L63 N.lugens is not distributed in Europe and Africa.
"in all the rice growing regions of the world" → " South, East-South and East Asian region."
Reviewer 4 Report
Titles of Y-axis in Figure 1, 3, 10, and 11 are too close to the axis. Please try to fix it.
